# *Candida albicans* Potassium Transporters

**DOI:** 10.3390/ijms23094884

**Published:** 2022-04-28

**Authors:** Francisco J. Ruiz-Castilla, Francisco S. Ruiz Pérez, Laura Ramos-Moreno, José Ramos

**Affiliations:** Department of Agricultural Chemistry, Edaphology and Microbiology, University of Córdoba, 14071 Córdoba, Spain; a32rucaf@uco.es (F.J.R.-C.); franruiz1996@gmail.com (F.S.R.P.); lauraramosm89@gmail.com (L.R.-M.)

**Keywords:** *Candida albicans*, potassium homeostasis, potassium transporters, plasma membrane, organelles, potassium, pathogenicity

## Abstract

Potassium is basic for life. All living organisms require high amounts of intracellular potassium, which fulfils multiple functions. To reach efficient potassium homeostasis, eukaryotic cells have developed a complex and tightly regulated system of transporters present both in the plasma membrane and in the membranes of internal organelles that allow correct intracellular potassium content and distribution. We review the information available on the pathogenic yeast *Candida albicans.* While some of the plasma membrane potassium transporters are relatively well known and experimental data about their nature, function or regulation have been published, in the case of most of the transporters present in intracellular membranes, their existence and even function have just been deduced because of their homology with those present in other yeasts, such as *Saccharomyces cerevisiae.* Finally, we analyse the possible links between pathogenicity and potassium homeostasis. We comment on the possibility of using some of these transporters as tentative targets in the search for new antifungal drugs.

## 1. Introduction

Potassium is crucial for life. All living organisms accumulate high intracellular concentrations of this cation to fulfil multiple functions and, additionally, eukaryotic cells must correctly distribute the right intracellular amounts of the cation in order to keep potassium homeostasis. With this objective, cells have developed a complex and tightly regulated network of potassium transporters present in the plasma membrane and in the membrane of intracellular organelles that must work in a coordinated way [1,2,3,4,5,6,7].

The physiological functions associated with the potassium fluxes have been well studied in the model yeast *Saccharomyces cerevisiae*. This cation is important in the regulation of cell volume and intracellular pH, the maintenance of membrane potential, protein synthesis and enzyme activation [8]. Although intracellular concentrations of potassium in *S. cerevisiae* range from 100 to 300 mM, approximately, the different intracellular compartments keep different amounts of the cation [8,9].

*Candida albicans* is a yeast species that, similar to *S. cerevisiae*, belongs to the class of fungi called ascomycetes. It is naturally diploid with a 14.3 megabase (Mb) genome consisting of eight chromosomes and a GC content of 33.5% [10]. *C. albicans* genome was sequenced and published in 2004 [11] and it is available in the *Candida* Genome Database [12]. Although this microorganism is not restricted to diploid conditions, the genome of *C. albicans* is very flexible and can withstand a wide assortment of variations in a continuously changing environment [13,14,15]. *C.*
*albicans* is among the most prevalent fungal species of the human microbiota and asymptomatically colonises healthy individuals [16,17,18]. This yeast is transmitted vertically from mother to child, and infections arise predominantly from the endogenous microbiota rather than other sources [19,20]. In healthy humans, *C. albicans* is usually a harmless member of the native microbiota and asymptomatically colonises many niches, including the gastrointestinal tract, reproductive tract, mouth and skin [21,22,23,24]. However, under certain circumstances, *C. albicans* can cause infections that range from superficial infections of the skin to life-threatening systemic infections [17]. A striking feature of this organism is its ability to grow either as a unicellular budding yeast or in filamentous pseudohyphal and hyphal forms [25,26]. The hyphal form has been shown to be more invasive than the yeast form and, on the other hand, the smaller yeast form is believed to represent the form primarily involved in dissemination [27,28,29]. Several environmental conditions can trigger changes in *C. albicans* morphology, including host temperature, pH, nutrient availability, or quorum sensing mechanisms [30,31,32]. Although information related to the pathogenicity processes in this organism increases each year, unfortunately, there is also a steady increase in the number of recorded cases of candidiasis annually caused by the development of drug resistance [33,34,35,36,37]. In this sense, a recent publication reviewed membrane cation transport in *C. albicans* and promising targets for drug development from a general point of view from ammonium to iron or calcium, but the authors focussed only on plasma membrane transport [35].

*C. albicans* needs potassium to carry out different vital functions and, for example, must compete with their host cells for the necessary potassium, which is usually present at a few mmol/l concentrations in the host extracellular fluids [38]. How this pathogenic yeast reaches efficient potassium homeostasis is far to be completely understood. While some of the potassium transporters present in its membranes are relatively well-known and experimental data on their structure, function or regulation are available, in many other cases, their existence has been inferred simply from their homology to those present in other yeasts and they have not been studied at all (Figure 1 and Table 1).

In this work, we have reviewed the published information describing the potassium transport systems in *C. albicans*; we have searched the *Candida* Genome Database [12] looking for additional transporters involved in potassium homeostasis and the understudied; and, finally, we describe the possible links between potassium homeostasis and the pathogenicity processes in *C. albicans*. We conclude that most of the potassium transporters previously reported in other yeasts are also present in *C. albicans,* and that some of these proteins may be promising targets in the search for new antifungal compounds.

## 2. Potassium Transporters in the Plasma Membrane

A summary of the most relevant information on the different potassium transporters in *C. albicans* is described below. Table 1 includes information about name, function, mechanism of transport or homology with other genes. In *S. cerevisiae*, the potassium fluxes importantly depend on the activity of the proton ATPase Pma1, which extrudes protons in a primary active transport process directly coupled to ATP hydrolysis. Consequently, a membrane potential difference is generated driving potassium fluxes [8]. In *C. albicans,* a homologue of *Sc*Pma1 has been identified (orf 19.5383). It is 895 amino acids long and has a percentage of identity of 81.2%. The characteristics of the protein have not been studied in detail, but it is reasonable to think that it has a similar function in both yeasts.

### 2.1. Trk1

The Trk (Transport of K^+^) family of transporters comprises a series of proteins, is present in all yeasts and is responsible for potassium transport. *TRK1* was the first gene encoding a potassium transporter identified and characterised in non-animal eukaryotic cells [57,58,59]. *TRK1* (orf 19.600) encodes a 1056 amino acid plasma membrane-localised protein in *C. albicans* and has a 32.7% identity to *S. cerevisiae* Trk1p. 

Although Trk1 is one of the most characterised transporters in yeasts, the information about its structure has been obtained mainly through its study in *S. cerevisiae*. Sequence analysis of this potassium transporter revealed that Trk1 has probably evolved from ancestral K^+^ channels (such as KcsA from *Streptomyces lividans*) via gene duplication from a single monomer of a KcsA-like channel [60]. The structure of Trk1 in *S. cerevisiae* (*Sc*Trk1) consists of one single polypeptide chain with four domains (A, B, C and D) and each Trk1 domain corresponds to one potassium channel monomer. These four domains assemble around a central axis forming the pore [61]. In this way, a tetra-M1PM2 structure is formed where M corresponds to a hydrophobic segment and P to an α-helix that enters the membrane and connects the M segments. In addition to this structure, *Sc*Trk1 also has an intracellular part, which is mostly located between the first and second MPM structures, forming a long hydrophilic loop (LHL) [1,35,62,63]. 

Regarding the transport mechanism mediated by Trk1, it is difficult to reach a clear conclusion, since it has not been possible to accurately measure the membrane potential in yeasts. However, there are specific data on the membrane potential of some filamentous fungi, which, applied to the Trk1 transport mechanism, suggest that it would function as a potassium uniporter [8,64,65]. Additionally, an electrophysiological approach in *S. cerevisiae* also revealed a secondary function for Trk proteins: chloride ion efflux [66]. However, this possible function seems to be poorly conserved between *S. cerevisiae* and *C. albicans*, being highly variant with respect to activation velocity, amplitude, flickering (channel-like) behaviour, pH dependence and inhibitors sensitivity [67].

It has been proposed that *TRK1* is an essential gene in *C. albicans*. This conclusion was reached after failing to obtain homozygous mutants for this gene using disruption cassettes [68]. This proposal is somehow surprising since *TRK1* is not essential in other yeast species. In *S. cerevisiae*, carrying two *TRK* genes [69] or even in *C. glabrata*, with only a single plasma membrane potassium transporter (*TRK1*), the mutants are viable [70]. Although the deletion of the gene increased the potassium requirements of the mutants, they were perfectly viable at high extracellular potassium [69,70,71]. In addition, a later study, in which the essentiality of genes was analysed using in vivo transposons, determined that *TRK1* was not an essential gene in *C. albicans* [72].

The existence of the *C. albicans* SN250 strain carrying the pseudogene *ACU1* (non-functional) and the *HAK1* gene disrupted [73] has allowed the study of Trk1 functions in the absence of additional plasma membrane potassium uptake systems. The study of that strain led to the conclusion that Trk1 behaves as a housekeeping transporter, and it is sufficient to supply the necessary potassium to *C. albicans* cells since they can grow at low potassium levels and can transport the cation with high affinity [41]. Furthermore, a different approach using heterologous expression of this gene was followed to obtain information about the transporter. The *CaTRK1* gene was expressed in the BYT12 mutant strain of *S. cerevisiae* (*trk1*∆*trk2*∆). *Ca*Trk1 was properly recognised and secreted to the plasma membrane and cells showed enhanced ability to grow at low potassium levels. The heterologous expression of the transporter conferred, in addition, tolerance to toxic cations such as Na^+^ or Li^+^ [38,39].

Studies on the transcriptional regulation of *CaTRK1* indicated that the gene is not importantly regulated at this level. This conclusion fits with previous reports on other yeast species [8,39,41,71].

### 2.2. Acu1

Acu (Alkali Cation Uptake) is a family of transporters located in the plasma membrane. These transporters are present in very few species of yeast, such as *Ustilago maydis*, *Pichia sorbitophila* or *C. albicans* [38,40]. Unfortunately, this reason makes these transporters not very well characterised. Although two genes of the *ACU* family (*ACU1* and *ACU2*) have been identified in fungi [40], only the *ACU1* gene was found in *C. albicans*. Interestingly, *C. albicans* seems to be the only *Candida* species carrying a gene encoding the Acu1 P-type ATPase. The first *C. albicans* strain sequenced, *SC5314* [11], is usually used as a wild type. However, it has been demonstrated that the putative *ACU1* gene is interrupted by a point mutation changing its codon 356 into a STOP codon [38,40]. The complete gene comprises 3243 bp and is divided into two orfs (19.2553 and 19.2552) by an intergenic region and apparently does not contain introns [40]. This fact should be kept in mind when using this strain to study potassium transport processes.

Acu proteins work as ATPases that drive K^+^ or Na^+^ into the cells. When the *Candida* gene was restored and expressed in *S. cerevisiae*, it was observed that it enhanced the ability of BYT12 cells (*trk1*∆*trk2*∆) to grow under potassium-limiting conditions and to transport this cation. By contrast, this transporter did not import sodium cations into *S. cerevisiae* cells [38]. Furthermore, BYT12 cells expressing *CaACU1* were able to increase the tolerance of yeast cells to toxic cations such as Li^+^ [38] or Na^+^ [39].

Regarding the transcriptional regulation of *ACU1* in *C. albicans*, it has been found that this gene is significantly upregulated by K^+^ starvation. In fact, after one hour in a medium containing low potassium, *ACU1* transcript levels increased more than one hundred times [39,41].

### 2.3. Hak1

The transporters belonging to the Hak family in yeasts (High-Affinity K^+^ transporter) are transporters located in the plasma membrane. These transporters are found in many organisms such as fungi or plants [74,75,76]. This transporter was first discovered in the yeast *Schwanniomyces occidentalis* by homology with the KUP system of *E. coli,* and it was named HAK due to its high affinity for potassium [42]. Subsequently, new orthologues were identified in other yeast species such as: *P. sorbitophila* [40], *D. hansenii* [77], *Hansenula polymorpha* [78] and *C. albicans* [38]. Interestingly no members of the gene family are present in the model yeast species *S. cerevisiae* or *Schizosaccharomyces pombe* [74].

*HAK1* (orf 19.6249) encodes a protein of 808 amino acids in *C. albicans*. The protein is very similar to Hak1 from *S. occidentalis* (56.35%), *D. hansenii* (52.14%) and *H. polymorpha* (37.91%). The transport mechanism of Hak1 was first described in the yeast *S. occidentalis* and in the fungus *Neurospora crassa*, where it was observed that this transporter functioned as a K^+^: H^+^ symporter [42,75]. There is not much information about the molecular organisation of this transporter [35]. A study reports on the structure and functioning of the KimA homologous transporter of *Bacillus subtilis* belonging to the KUP family and identified key residues for K^+^ and proton binding, which are conserved in KUP proteins [79].

As mentioned above, a mutant lacking functional *HAK1* was obtained by using disruption cassettes in *C. albicans*. Although no phenotypes related to potassium homeostasis were originally studied, the mutant showed a defect in infectivity but no effects on morphogenesis or proliferation [73]. Later on, a detailed study of the mutant indicated that this transporter may function as a symporter with protons. *Ca*Hak1 is probably important during growth at low potassium and/or at acidic pH. Furthermore, Hak1 contributes to avoiding lithium toxicity at low pH [41]. Heterologous expression of *CaHAK1* in the *S. cerevisiae* strain BYT12 (*trk1*∆*trk2*∆) demonstrated its role in potassium transport since an improvement in growth at limiting potassium and in rubidium transport (Rb^+^ is used as a K^+^ analogue in the kinetics characterization) was determined [38,39]. In addition, the cells that expressed *CaHAK1* improved their tolerance to sodium [39]. 

*HAK1* transcriptional regulation in *C. albicans* seems to be strongly induced in response to potassium starvation [39,41]. This behaviour has also been observed in other yeasts [65,78,80].

### 2.4. Tok1

Tok1 (Transport Outward Potassium) has been described as a plasma membrane potassium channel that regulates specific potassium efflux in yeast. *TOK1* (orf 19.4175) encodes a protein of 741 amino acids in *C. albicans* with a 32.1% percent of identity to Tok1 from *S. cerevisiae* (also named Ypk1, Duk1, Ykc1 or YORK)

Most of the known information on this transporter is inferred from what is known in the model yeast *S. cerevisiae*. However, this protein has hardly been studied in *C. albicans*. 

Tok1 structure in *S. cerevisiae* has been studied in detail. This protein contains two pore-forming P domains, structural entities that are common to all known K^+^-selective ion channels. Among proteins encoded in the yeast genome, only the Tok1 channel contains P domains [81,82]. The main function of the channel is K^+^ efflux. *Sc*Tok1 is regulated by the membrane potential and the extracellular potassium, so that the depolarization of the membrane causes the opening of the channel and with it, the exit of potassium to the cell exterior. A secondary function of Tok1 has been proposed. The proposal was that Tok1 can mediate K^+^ influx under specific conditions [83,84].

Yeast strains carrying null mutations in *TOK1* are viable in *S. cerevisiae* and *C. albicans* [43]. In *S. cerevisiae*, deletion of the *TOK1* gene results in significant plasma membrane depolarization, whereas strains overexpressing the *TOK1* gene are hyperpolarised. Therefore, it has been proposed that this transporter may have the function of allowing yeast to maintain the plasma membrane potential under some stress conditions [85]. On the other hand, the function of Tok1 has not been studied in detail in *C. albicans*. A published article demonstrated that deletion of *TOK1* in *C. albicans* completely abolishes the currents and gating events characteristic of Tok1p and suggests that Tok1 modulates sensitivity to human salivary histatin 5 [43].

### 2.5. Cnh1

The *CNH1* gene (orf 19.367) encodes a protein of 800 amino acids long located in the plasma membrane of *C. albicans*. This protein exhibits a high level of similarity in the sequence, size, structure, and functional domains with the Na^+^/H^+^ antiporters of fungi [2,44,86]. *CaCNH1* gene is a homologue of *NHA1* from *S. cerevisiae* with a percent of the identity of 41.2% and encodes a Na^+^/H^+^ antiporter involved in the regulation of K^+^ and Na^+^ content and maintenance of intracellular pH [2,44,86].

A mutant of *CaCNH1* was created using a cassette to generate a gene-disruption construct for sequential deletion of both copies of the *CNH1* gene. Using this mutant, the authors demonstrated that complete deletion of the gene did not change cellular tolerance to high concentrations of NaCl or LiCl [44]. Later on, and using a different genetic background, new mutants in the gene were obtained. With this purpose, a two-step procedure to delete the gene using disruption cassettes was followed. Results confirmed that the deletion of *CNH1* did not affect sodium or lithium tolerance. Moreover, growth in CsCl was not perturbed but the deletion resulted in increased sensitivity to high external concentrations of KCl and RbCl (Rb^+^ is used as a K^+^ analogue in the kinetics characterization) in a pH-dependent manner, which corresponds to the nature of an antiport mechanism using the gradient of protons across the plasma membrane. The sensitivity of the *cnh1* mutant to external KCl was caused by a lower K^+^ efflux. Furthermore, the reintegration of the *CNH1* gene into the *cnh1* null mutant restored its potassium and rubidium tolerance [45]. Additionally, the *Ca*Cnh1 antiporter was also heterologously expressed in the *S. cerevisiae* mutant strain BW31 (*ena1-4*Δ*nha1*Δ), resulting in increased tolerance to sodium, lithium, potassium, and rubidium cations. Taken together, the available data revealed that *Ca*Cnh1 plays an important role in *C. albicans*, ensuring tolerance to potassium and rubidium and participating in the regulation of intracellular potassium content [45]. In addition, it has been suggested that the physiological role of *Ca*Cnh1 in *C. albicans* could be much broader than simple detoxification from surplus alkali cations, and regulation of intracellular pH can be one of these functions [45].

A schematic representation of the putative secondary structure and topology of Trk1, Acu1, Hak1 and Cnh1 has been recently proposed [35].

### 2.6. Ena21-22

The ENA (Efflux of Natrium) family belongs to the category of P type-ATPases, and its members are found exclusively in fungi, bryophyta and protozoa. The existence of the ENA ATPases in almost all fungi, as well as in bryophytes and protozoa, suggests that these proteins are required for the adaptation to living conditions that prevail in organisms with very different lifestyles [74,87]. Two *ENA* genes have been found in *C. albicans*: *ENA21* (orf19.5170) and *ENA22* (orf19.6070). *ENA21* encodes a protein with 971 amino acids long while *ENA22* encodes a protein with 1067 amino acids long. The *C. albicans ENA21* gene and its paralogue *ENA22* are orthologues of the *S. cerevisiae ENA*2 gene that encodes a P-type ATPase sodium pump [46]. The percent identity of *Ca*Ena21p and *Ca*Ena22p with *Sc*Ena2p is 50.8% and 52.9%, respectively.

*ENA* family of genes in *S. cerevisiae* is composed of 4–5 genes [87], but most of the information reported has been obtained by studying Ena1. These ATPases are composed of ten transmembrane fragments and two cytoplasmic loops that play a central role in the functional mechanism of the enzyme [87]. The main function of the *Sc*Ena ATPases is to mediate the effluxes of Na^+^ or K^+^ in order to control cation contents and cytosolic pH [87]. They are involved in sodium tolerance but they are not specific for Na^+^ (or Li^+^) extrusion. These ATPases also transport K^+^ and contribute to regulating intracellular concentrations of the cation as deduced from the characterization of the Ena1 ATPase activity in *S. cerevisiae* [2,88].

Almost nothing is known about the physiological functions of Ena21-22 in *C. albicans,* but the strong induction of Ena21 in response to stress situations has been reported, and it has been proposed that it works as a Na^+^ ion transporter [46]. It is obvious that much more work is required to understand the function/s of *Ca*Ena pumps, but it seems conceivable that they also may transport K^+^ as happens in *S. cerevisiae*.

### 2.7. Kch1

*KCH* (Potassium regulator of CcH1) has been identified as a family of genes involved in the *S. cerevisiae* low-affinity K^+^ transport process but the information available is confusing. At least two genes, *KCH1* and *KCH2,* belong to the family and the corresponding proteins are proposed to be located in the plasma membrane [89]. Although Kch1 and Kch2 were first proposed to be putative low-affinity potassium transporters, this possibility was not confirmed later and their role in potassium homeostasis may be indirect. 

*C. albicans* carries only one homologue of the family: *KCH1* (orf 19.6563) [47]. This gene encodes a protein of 659 amino acids in length with a percentage of identity of 17.0% with Kch1 of *S. cerevisiae*. A mutant of this transporter was obtained using disruption cassettes in *C. albicans* [47]. Apparently, Kch proteins do not exert identical functions in *S. cerevisiae* and in *C. albicans*. Characterization of *S. cerevisiae* mutants lacking *KCH1* and *KCH2* showed that they are smaller and hyperpolarised compared to wild-type cells, they grow better under limiting potassium concentrations, and they exhibit altered growth in the presence of monovalent cations [48]. On the contrary, the deletion of *KCH1* in *C. albicans* did not affect cell growth under potassium-limiting conditions, their tolerance to these monovalent cations, or their membrane potential. Therefore, it was proposed that *Kch*1 does not participate in the regulation of monovalent cation homeostasis in *C. albicans* [48].

## 3. Intracellular Potassium Transporters

None of the putative potassium transporters or proteins involved in potassium homeostasis present in the membranes of intracellular organelles have been studied in detail in *C. albicans.* Experimental data on their function and role in global cation homeostasis are not available since most of these carriers have been identified based on database searches and their homology to those present in other yeast species. 

### 3.1. Nhx1 (Na^+^/H^+^ Exchanger)

In *C. albicans*, a transporter homologous to Nhx1 of *S. cerevisiae* is present. The protein is 58.5% identical to *Sc*Nhx1. *NHX1* (orf 19.4201) encodes a protein 663 amino acids long in *C. albicans*, which may work as a Na^+^/H^+^ (or K^+^/H^+^) antiporter. In *S. cerevisiae*, this transporter is located in the vacuole and late endosomal compartments. It appears that Nhx1 may be involved in different functions, such as intracellular sodium sequestration, luminal and cytoplasmic pH regulation and vacuolar trafficking [49,90,91]. As deduced from the similarity between the proteins in *Candida* and *Saccharomyces,* it is conceivable to propose related functions in both yeasts.

### 3.2. Vhc1 (Vacuolar Protein Homologous to CCC Family)

An ortholog of the *S. cerevisiae* Vhc1 transporter has been found in *C. albicans* (orf 19.6832, C3_06710W). The gene encodes a protein of 663 amino acids with a percentage of identity of 12.9% with respect to *Sc*Vhc1. Although nothing is known about the function of the protein in *C. albicans*, in *S. cerevisiae,* the transporter is located in the membrane of the vacuole, participates in the regulation of cation content and vacuolar morphology, and has been proposed to work as a K^+^/Cl^−^ cotransporter [50,51,92].

### 3.3. Vnx1 (Vacuolar Na^+^/H^+^ Exchanger)

In *S. cerevisiae*, Vnx1 works as an antiporter that uses the proton gradient generated by the Vma1 H^+^-ATPase to mediate the transport of potassium (or Na^+^) into the vacuole, thus contributing to the regulation of cytosolic pH [52]. An ortholog of this transporter has been found in *C. albicans* (orf 19.7670, CR_10800C). This gene encodes a protein of 923 amino acids homologous to the vacuolar *Sc*Vnx1 (40.8%). A homozygous null mutant lacking this gene was constructed using a PCR-based gene disruption technique in *C. albicans*. The mutant showed reduced capacity to damage oral epithelial cells, and the same work reported that the gene is upregulated during oral infection [53]; although, nothing has been published in relation to potassium homeostasis. 

### 3.4. Vcx1 (Vacuolar Ca^2+^/H^+^ Exchanger)

The main activity of the protein encoded by *VCX1* in *S. cerevisiae* may be vacuolar Ca^2+/^H^+^ exchange; although, it may play a role in potassium transport [54]. *VCX1* encodes (orf 19.405) a protein with a length of 416 amino acids in *C. albicans*. This protein has a percentage of identity of 59.3% in comparison with *Sc*Vcx1 and nothing else is known about its putative functions in Ca^2+^ or K^+^ fluxes.

### 3.5. KHE (K^+^/H^+^ Exchanger)

The possible potassium transporters in the membrane of the mitochondria are not well identified. In *S. cerevisiae,* three genes related to mitochondrial potassium homeostasis have been identified (*MDM38*, *MRS7* and *YDL183C*) and they may be involved in potassium transport [52,93,94,95,96]. Searches in the *Candida* database indicate the existence of genes homologous to *MRS7* and *YDL183C*. Heterozygous mutants of both genes were obtained, but not characterised, in the context of a screen to identify new genes necessary for filamentous growth [97]. *MRS7* (orf 19.3321) would encode a protein present in the membrane of *C. albicans* mitochondria [55]. The putative protein would be 508 amino acids long; 40.7% identical to the corresponding protein in *S. cerevisiae.* Finally, *C3_01680C* (orf 19.1676) would encode a protein 369 amino acids long; 26.0% identical to the corresponding protein in *S. cerevisiae* (Ydl183c). The only information available indicates that the protein is related to potassium fluxes [56].

## 4. Potassium Homeostasis and Pathogenicity

Invasive fungal infections have been increasing significantly in recent decades, contributing to high incidences and mortality in immunosuppressed patients [98,99,100]. One example is that annual treatment costs in the US are estimated at more than a billion dollars [101]. Therefore, it is urgent to find new targets to develop more selective antifungal drugs that target such an important pathogen as *C. albicans*. As mentioned above, potassium plays a vital role in living organisms, making this cation crucial in multiple physiological processes [8]. In addition, some publications have pointed out that new targets may be related to the yeast potassium homeostasis processes and to the transporters of the cation [33,35,102]. We summarise below the relevant information on the most promising targets related to potassium homeostasis for future drug development against *C. albicans*.

### 4.1. Pma1

Pma1 ATPase drives K^+^ fluxes and possesses important attributes that make it desirable as a target for antifungal drug discovery. This protein is essential for the physiology of fungal cells, being necessary for the formation of a large electrochemical proton gradient and the maintenance of intracellular pH. As mentioned above, *Ca*Pma1 has not been well characterised, but it has been proposed that complete or partial inhibition of the proton pump may be lethal for that yeast. Omeprazole inhibits the growth of *S. cerevisiae* and *C. albicans* in a pH-dependent manner. Studies with this substance have indicated that its action is closely correlated with inhibition of the H^+^-ATPase, and a region of the ‘proton pump’ has been proposed to be valuable as a potential interaction domain for antifungal agents. These studies concluded that the H^+^-ATPase is a highly desirable target for the development of novel antifungal therapeutics [103,104]. Unfortunately, we have no new information or significant advances in the field more than two decades later, which reflects the complexity of the subject.

### 4.2. Trk1

Salivary histatins (Hsts) are structurally related histidine-rich cationic proteins produced by acinar cells in human salivary glands and are key components of the non-immune host defence system of the oral cavity [68,105,106]. Histatin 5 (Hst 5) kills *C. albicans* via a multistep process, which includes binding to the *Candida* surface protein Ssa1/2 and seems to require the Trk1 potassium transporter [68]. The information about this exciting finding is limited and deserves more research, but it is clear that *Ca*Trk1 is an important effector for Hst 5 in *C. albicans*. Moreover, it has also been proposed that DIDS (anion channel inhibitor) blocks native chloride conductance most probably mediated by *Ca*Trk1 [68].

The importance of the Trk1-mediated potassium uptake as a target for antifungals has been studied in two *Candida* species carrying only one plasma membrane potassium uptake system (Trk1). In one report, the Trk1 potassium transporter in *C. krusei* and *C. glabrata* was suggested to serve as a target for the development of new antifungal drugs [107]. Later work in *C. glabrata* demonstrated that the loss of Trk1 resulted in diminished virulence as assessed by two insect host models, *Drosophila melanogaster* and *Galleria mellonella*, and experiments with macrophages. Macrophages killed *trk1*Δ cells more effectively than wild-type cells, but macrophages accrue less damage when co-cultured with *trk1*Δ mutant cells compared to wild-type cells. Moreover, the same work showed that low levels of potassium in the environment increased the adherence of *C. glabrata* cells to polystyrene and the propensity of *C. glabrata* cells to form biofilms, which is also related to the pathogenicity processes [108]. In summary, Trk1 is not an essential protein for *C. albicans,* but defective Trk1 proteins may have consequences in the pathogenic process.

### 4.3. Tok1

As already mentioned, Tok1 works as an outward rectifier potassium channel in yeast. *Ca*Tok1 has been proposed to be a promising target for antifungal drugs based on results obtained with the toxin Hst 5, which induces a high efflux of K^+^. Although, once again, more work is required to reach definitive conclusions, the involvement of the channel in the viability of *C. albicans* after Hst 5 treatment has been shown, and, moreover, deletion of one or both alleles proportionally increased resistance to the toxin [43].

### 4.4. ENA

The ENA system, involved in K^+^ or Na^+^ extrusion, does not have a functional homologue in animal cells; although, we found two human proteins that were about 20% homologous with *Ca*Ena (Table 1). For this reason, this ATPase is a potential target for antifungal and antiparasitic drugs or herbicides [87,109,110]. In *C. albicans,* the activities of the Ena21-22 system must be elucidated in detail, but it may be a convenient drug target as suggested in *S. cerevisiae* or in *C. glabrata* [87,111].

Even less information is available about the potential use of additional transporters such as Cnh1 or Kch1 in the search for drug development [35,47]. In conclusion, although some of the mentioned transporters may be very promising targets, further work on the structure of these proteins should clarify their potential use in the search for improved pharmacological strategies.

## 5. Conclusions and Perspectives

Our knowledge regarding potassium homeostasis and fluxes in *C. albicans* has significantly increased in the last years, but it is still fragmentary and uneven. The function, possible working mechanism or regulation have been analysed in some plasma membrane transporters, but, in some others, the only information available is indirect or it has been deduced from their homologous genes in other fungi, mainly, *S. cerevisiae*. Moreover, no real advances have been made in understanding putative potassium transporters present in intracellular organelles since specific experimental data have not been provided. Additionally, the possible differences in expression or activity of the transporters when *C. albicans* grows in the cell or in the hyphal form have not been approached and this is a fully open field. 

Although some works report the use of homozygous mutants, an important effort is required to adapt, improve and develop friendly molecular techniques that will help to prepare mutants and will provide relevant information about the roles of the different transporters under different ecological conditions. It would be of utmost importance to obtain a global idea of how *Candida* cells reach efficient potassium homeostasis through the tight regulation of the different transporters working in a coordinated way.

Globally speaking, one of the most interesting aspects of this field is the relationship between potassium homeostasis and pathogenicity. Several transporters have been postulated as possible targets in the search for new antifungal drugs; however, this attractive picture lacks structural and molecular details that are extremely important to succeed in this search. The existence of three types of K^+^ influx systems in the plasma membrane of *C. albicans* may be a drawback when designing strategies in the search for new antifungal drugs, since, very probably, specifically inhibiting one of the three different proteins may not be enough to kill the pathogen. Similarly, although inhibiting potassium extrusion through Tok1 or Ena proteins is an additional possibility, once again, other transporters such as Cnh1 may supply an alternative potassium extrusion pathway. From this point of view, in our opinion, Vnx1 or Cnh1 could be attractive and more realistic possible targets. There are no homologous genes to *VNX1* or *CNH1* in humans and, in fact, in addition to their function as potassium transporters, they are most likely involved in pH maintenance, which can be an additional advantage.

## Figures and Tables

**Figure 1 ijms-23-04884-f001:**
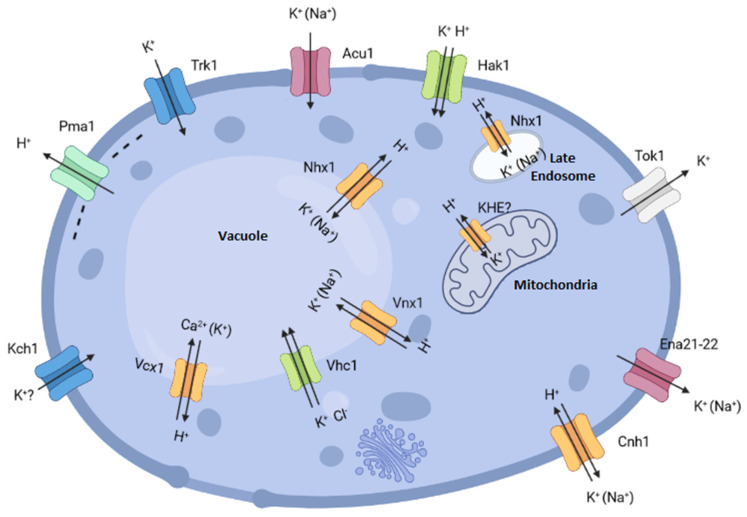
Putative potassium transporters identified in *C. albicans***.** The ATPase encoded by *PMA1* extrudes protons and creates a membrane potential that drives K^+^ movements. The function of the KHE system and Kch1 transporter has not been definitively determined in either *S. cerevisiae* or *C. albicans*.

**Table 1 ijms-23-04884-t001:** Potassium transporters in *C. albicans*.

Name	Orf	Possible Function	Proposed Mechanism	Chromosome	Homology	Specific Experimental Data	Relevant References
Trk1	19.600	K^+^ influx	K^+^ Uniporter	R	*Sc*Trk1 (32.7%) *Sp*Trk1 (28.7%)	Yes	[38,39]
Acu1	19.2553 and 19.2552	K^+^ (Na^+^) influx	K^+^ (Na^+^) ATPase	R	- *	Yes	[38,39,40]
Hak1	19.6249	K^+^ influx	K^+^: H^+^ Symporter	1	*Sch*Hak1 (56.35%) *D*hak1 (52.14 %) *Hp*Hak1 (37.91%)	Yes	[38,39,41,42]
Tok1	19.4175	K^+^ efflux	K^+^ efflux channel	4	*Sc*Tok1 (32.1%) *Hs*Kcnk9 (11.3%)	Yes	[43]
Cnh1	19.367	K^+^ (Na^+^) efflux. Maintenance of intracellular pH	K^+^(Na^+^)/H^+^ antiporter	4	*Sc*Nha1 (41.2%) *Sp*Sod22 (37.1%)	Yes	[44,45]
Ena21	19.6070	K^+^ (Na^+^) efflux	P-type ATPase	1	*Sc*Ena2 (50.8%) *Sp*Cta3 (44.4%)	No	[46]
Ena22	19.5170	K^+^ (Na^+^) efflux	P-type ATPase	7	*Sc*Ena2 (52.9%) *Sp*Cta3 (45.9%)	No	[46]
Kch1	19.6563	Unknown	Unknown	7	*Sc*Kch1 (17.0%)	Yes	[47,48]
Nhx1	19.4201	Intracellular K^+^ (Na^+^) sequestration	K^+^ (Na^+^)/H^+^ exchanger	6	*Sc*Nhx1 (58.5%) *Sp*Cpa1 (46.0%) *HsSlc9a9* (25.6%)	No	[49]
Vhc1	19.6832	Vacuolar cation content and morphology	K^+^-Cl^−^ cotransporter	3	*Sc*Vhc1 (12.9%) *Sp*Vhc1 (9.1%) *HsSlc12a9* (3.8%)	No	[50,51]
Vnx1	19.7670	Vacuolar K^+^ (Na^+^) content and regulation of cytosolic pH	K^+^ (Na^+^)/H^+^ exchanger	R	*Sc*Vnx1 (40.8%) *Sp*Sst1 (33.8%)	No	[52,53]
Vcx1	19.405	Regulation of vacuolar Ca^2+^ and K^+^ content	Ca^2+^ (K^+^)/H^+^ exchanger	1	*Sc*Vcx1 (59.3%) *Sp*Vcx1 (47.2%)	No	[54]
Mrs7 (KHE)	19.3321	Involved in mitochondrial K^+^ homeostasis	Unknown	1	*Sc*Mrs7 (40.7%) *Sp*Mdm28 (37.5%) *Hs*Letm1 (20.8%)	No	[52,55]
C3_01680C (KHE)	19.1676	Involved in mitochondrial K^+^ homeostasis	Unknown	3	*Sc*Ydl183c (26.0%) *Sp*SPAC23H3.12c (17.8%)	No	[52,56]

* The *ACU1* gene in the strain SC5314 is interrupted by a stop codon in position 356 and it appears as two different orfs in the Candida Genome Database. Sc *Saccharomyces cerevisiae*. Sp *Schizosaccharomyces pombe.* Sch *Schwanniomyces occidentalis*. Dh *Debayromyces hansenii*. Hp *Hansenula polymorpha*. Hs *Homo sapiens*.

## Data Availability

Not applicable.

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
