# Peer review of "Candida albicans Potassium Transporters"

_ijms, 2022, doi:10.3390/ijms23094884_

Round 1

Reviewer 1 Report

It's a useful review. I made comments in the manuscript (attached). The major comments related to:

  1. improving Table 1 by including human homology and cellular localization. Does human homology or cellular localization impact the utility of the gene as a target? This table should be moved up in the MS proximal to (and sort of integrated with) Fig. 1.
  2. Fig. 2 seems unnecessary or redundant and could be incorporated into Fig. 1.
  3. Overall, language is ok, but there was one type and some wording suggestions were made in a few places.
  4. several suggestions for discussion (see comments in MS).

Author Response

Minor comments ( mostly grammar or typos) have been corrected

Table 1 has been modified, moved to Section 2 and human and pombe homologies have been included

As suggested, Figure 2 has been cleared (and Figure 1 has been slightly modified to include Pma1 ATPase)

Sections 4 and 5 have been improved according to the suggestions

Reviewer 2 Report

In this review, the authors list the information available on the proteins that could act as potassium transporters in the plasmatic membrane or organelle membranes of Candida albicans. The text is well written and is, in my opinion, interesting for IJMS. I only have minor corrections, which can be found in the pdf version of the manuscript attached to this report.

Author Response

All minor comments and changes have been accepted and some sentences have been rewritten (see changes in the new version of the manuscript).

Figure 1 has been modified and Nhx1 has been included as a potential vacuolar membrane protein.

Table 1 has been moved to the begining of Section 2 as suggested.

Reviewer 3 Report

The manuscript entitled Candida albicans potassium transportersby Francisco J. Ruiz-Castilla et al. discussed the information about potassium transporter available in the pathogenic yeast Candida albicans. Manuscript is nicely written, however, it can be improved after revision.

  1. The author must compare similar review work done earlier with this new work in the introduction. Emphasis about the difference in their work in the motivation part.
  2. Author must revise abstract as still some important information is missing for readers. Many sentences are not clear and complete.
  3. I recommend author to include qualitative work done previously to include in revised version. Author can include description and figures for the same.

Author Response

Thank you for the comments that we appreciate very much.

1. The author must compare similar review work done earlier with this new work in the introduction. Emphasis about the difference in their work in the motivation part.

We have taken into consideration this suggestion and we have added some comments in the Introduction section, specially in relation to a recent review by Volkova et al (2021).

2. Author must revise abstract as still some important information is missing for readers. Many sentences are not clear and complete.

Abstract has been reviewed and corrected.

3. I recommend author to include qualitative work done previously to include in revised version. Author can include description and figures for the same.

We do not see the point raised by the reviewer since the whole review has been prepared on the basis of previous work. Figure 1 and table 1 are perfect examples since they include all potassium transporters so far identified.